# Study on Solidification and Stabilization of Antimony-Containing Tailings with Metallurgical Slag-Based Binders

**DOI:** 10.3390/ma15051780

**Published:** 2022-02-26

**Authors:** Yunyun Li, Wen Ni, Wei Gao, Siqi Zhang, Pingfeng Fu, Yue Li

**Affiliations:** 1School of Civil and Resource Engineering, University of Science and Technology Beijing, Beijing 100083, China; lyyustb@163.com (Y.L.); zsq2017@ustb.edu.cn (S.Z.); pffu@ces.ustb.edu.cn (P.F.); 18532334075@163.com (Y.L.); 2Key Laboratory of Resource-Oriented Treatment of Industrial Pollutants, University of Science and Technology Beijing, Beijing 10083, China; 15201455635@163.com; 3Key Laboratory of High-Efficient Mining and Safety of Metal Mines, Ministry of Education, University of Science and Technology Beijing, Beijing 10083, China; 4School of Energy and Environmental Engineering, University of Science and Technology Beijing, Beijing 100083, China

**Keywords:** metallurgical slag, solidification/stabilization, antimony, steel slag, mine tailings

## Abstract

Blast furnace slag (BFS), steel slag (SS), and flue gas desulfurized gypsum (FGDG) were used to prepare metallurgical slag-based binder (MSB), which was afterwards mixed with high-antimony-containing mine tailings to form green mining fill samples (MBTs) for Sb solidification/stabilization (S/S). Results showed that all MBT samples met the requirement for mining backfills. In particular, the unconfined compressive strength of MBTs increased with the curing time, exceeding that of ordinary Portland cement (OPC). Moreover, MBTs exhibited the better antimony solidifying properties, and their immobilization efficiency could reach 99%, as compared to that of OPC. KSb(OH)_6_ was used to prepare pure MSB paste for solidifying mechanism analysis. Characteristics of metallurgical slag-based binder (MSB) solidified/stabilized antimony (Sb) were investigated via X-ray diffraction (XRD), field emission scanning electron microscopy (SEM), energy dispersive spectroscopy (EDS), Fourier transform infrared spectroscopy (FT-IR), and X-ray photoelectron spectroscopy (XPS). According to the results, the main hydration products of MSB were C-S-H gel and ettringite. Among them, C-S-H gel had an obvious adsorption and physical sealing effect on Sb, and the incorporation of Sb would reduce the degree of C-S-H gel polymerization. Besides, ettringite was found to exert little impact on the solidification and stabilization of Sb. However, due to the complex composition of MSB, it was hard to conclude whether Sb entered the ettringite lattice.

## 1. Introduction

In 2018, China produced 12.11 billion tons of tailings, accounting for 35.11% of the total production of bulk industrial solid waste. The tailings after mineral processing are usually discarded as solid waste in tailings ponds without any treatment, carrying potential heavy metal pollution risks [1]. Antimony is a toxic metalloid in the group VA of the fifth period of the periodic table. Antimony used in the lead-zinc tailings is mainly stibnite (Sb_2_S_3_) [2], and many studies reported that the migration and transformation of antimony is mainly Sb(OH)_6_^−^ [3,4]. In this respect, a series of reactions such as oxidation and leaching occurs under the combined action of oxygen, water, and microorganisms, making Sb_2_S_3_ soluble SbO_3_^−^ and hydrolyze to Sb(OH)_6_^−^, and then attached to the surface of various solid particles in the tailings in the form of adsorption [3,5]. This part of antimony usually has high activity, and without proper pretreatment managed, it can cause serious or even catastrophic consequences.

For waste with increased heavy metal concentrations, a feasible management approach may be to landfill the waste [6,7]. At present, the use of cement as a binder for landfill is already a mature technology. Ordinary Portland cement (OPC) is traditionally used for landfill solidification/stabilization (S/S) and has long-term stability [8,9,10,11]. For example, there are research reports where Portland cement, fly ash, clay, gypsum, and blast furnace slag were used as solidification/stabilization materials to limit the leaching potential of antimony [12]. According to their results, the leaching rate of antimony-containing waste residue was low and fulfilled the landfill standard, and gypsum was successfully used in immobilizing the antimony. Cornelis investigated the leaching characteristic of antimony by mixing Portland cement with KSb(OH)_6_ [13]. They found that various hydration minerals of cement had limited effects on antimony, such as AFm minerals and C-S-H gel. Unfortunately, they did not do a mineralogical analysis to verify the solidifying mechanism of antimony. However, the production of cement is accompanied by the large CO_2_ emission, yielding the high cost and limiting the use of the final product [8]. To reduce carbon emissions and to make it possible to absorb a large amount of metallurgical solid waste, blast furnace slag (BFS), steel slag (SS), and flue gas desulfurized gypsum (FGDG) were proposed as cementing binders [14,15,16]. In the hydration process, BFS provides potentially active silicon–oxygen and aluminum–oxygen tetraheda, SS is a source of alkaline substances, and FGDG ensures sulfate ions [17,18]. This enables one to form ettringite with low solubility and amorphous C-S-H gels, which enhance the physical and mechanical properties of the cured material to stabilize Sb [13]. At present, most research aimed at evaluating the environmental friendliness of S/S materials and the leaching characteristics of heavy metals using different leaching methods [4,19]. According to our previous research, the metallurgical slag-cementing binders effectively limits the leaching concentration of As and Sb, and the dominant leaching mechanism for Sb is diffusion [14,15,20]. In addition, in our previous research, the solidification mechanism of As was confirmed [21,22]. Arsenic atom can replace the aluminum atom or silicon arsenate with lower solubility, and arsenic can be adsorbed and encapsulated by C-S-H gel [23]. However, as far as we know, there are no relative studies to elucidate the solidifying mechanism of high Sb concentration tailings with metallurgical-slag-based binders from the perspective of mineralogy.

In this study, high-Sb-containing mine tailings (Sb-MT) were immobilized using the total solid waste S/S materials (MSB). The leaching tests were carried out according to HJ 557-2010 Chinese standard to evaluate the immobilization efficiency of antimony and the antimony leaching trend. For better understanding of the immobilization mechanisms of antimony-containing tailings using metallurgical slag-cementing agent, their microscopic characterization was performed via Fourier transform infrared spectroscopy (FT-IR), X-ray photoelectron spectroscopy (XPS), and scanning electron microscopy paired with energy dispersive spectrometer (SEM-EDS).

## 2. Materials and Experimental Procedures

### 2.1. Materials

In this study, the materials included BFS, SS, FGDG, high Sb-containing mine tailings (Sb-MT), and ordinary Portland cement PO 42.5 (OPC). Sb-MT samples were purchased from a lead–zinc ore dressing plant in Hechi Nandan, Guangxi province, China. The BFS, SS, and FGDG were provided by Jintaicheng Environmental Resources Co., Ltd., Xingtai, China. The chemical compositions and Blaine fineness values of these materials are given in Table 1. The leaching concentration of antimony in the raw materials was analyzed via inductively coupled plasma atomic emission spectroscopy (ICP-AES). No heavy metals were detected except Sb-MT, with a leaching Sb content of 0.524 mg/L. The XRD pattern of Sb-MT, BFS, SS, and FGDG are shown in Figure 1, revealing the presence of quartz, calcite, and fluorite as the primary phases in the Sb-MT. BFS was mainly in the glassy state, being composed of SiO_2_, CaO, and Al_2_O_3_. The primary phases of SS were SiO_2_, CaO, Fe_2_O_3_, and Al_2_O_3_, as well as ꞵ–C_2_S, C_3_S, C_3_A, and RO phases. The major composition of the FGDG was CaSO_4_∙2H_2_O.

### 2.2. Mixture Proportion of Mining Fill Samples

The relative weight percentages of the cementitious materials correspond to those from Ref. [22] and are given in Table 2. OPC was used as cementitious material for the control group treatment. Slump tests were completed according to GB/T 2419-2005 Chinese standard. The slumps of MBT and OPC were both 300 mm, respectively, indicating their good workability.

### 2.3. Experimental Procedures

The MBT pastes were poured into 50 × 50 × 50 mm^3^ steel molds, and then cured in a curing chamber (at the relative humidity of approximately 90% and 40 °C, mimicking the conditions of underground mine storage in Guangxi) [22]. The unconfined compressive strength (UCS) of the specimens was assessed as according to GB/T17671-1999 standard at curing times of 3 d, 7 d, 28 d, and 90 d.

Toxicity characteristic leaching tests were conducted via the horizontal vibration method using MBT and OPC at curing times of 3 d, 7 d, 28 d, and 90 d [21].

MSBs were prepared using a 300 mg/L KSb(OH)_6_ solution to investigate the hydration and curing mechanisms. All the samples were crushed and terminated with pure ethyl alcohol after 3 d, 7 d, and 28 d, respectively.

X-ray diffraction (XRD) was identified the pastes of the hydration products. XRD analysis was performed on an Ultima IV X-ray diffractometer (Rigaku Mechatronics Co., Ltd, Akishima-shi, Tokyo, Janpan) with a copper Kα radiation source (λ = 1.5406 Å) operating at 30 mA and 50 kV. The scan step was 0.02°, and the scan 2θ was from 5° to 70°.

The microstructure of hydrated pastes was observed via field emission scanning electron microscope (SUPRA 55,Carl Zeiss, Oberkochen, Germany). The working voltage was 10 kV, at a vacuum level lower than 9.9 × 10^−6^ mbar. The chemical compositions of hydration products were analyzed using an energy dispersive X-ray spectrometer (EDX, Carl Zeiss, Oberkochen, Germany) coupled with FE-SEM.

Structural and chemical bond characterization was performed via Fourier transform infrared spectroscopy (FT-IR). The FT-IR spectra of the samples were recorded in 350–4000 cm^−1^ using a NICOLET470 infrared spectrometer(ThermoFisher, Maltham, MassachusettsUS), at a sensitivity of 4 cm^−1^.

The chemical states of the elements were used X-ray photoelectron spectroscopy (XPS). The measurements were done at room temperature under a vacuum of 7 × 10^−9^ mbar by means of an XPS system (AXIS ULTRA^DLD^, kratos Analytical Ltd., Kyoto, Japan) using a nonmonochromatic Al Kα radiation source (1486.6 eV). The scanned surface area was 700 μm × 300 μm. During the XPS experiments, the C1s signal of adventitious hydrocarbons (284.8 eV) serves as the reference line.

## 3. Results

### 3.1. UCS Test Results

Figure 2 displays the unconfined compressive strengths of MBT and OPC with increasing curing time. In both cases, the UCS values were above 1 MPa, meaning that the samples satisfied the requirements for mining backfills. Meanwhile, the unconfined compressive strength of MBT was higher than that of OPC, indicating that metallurgical slag-cementing agent had good mechanical performance and broad application prospects.

### 3.2. Toxicity Characteristic Leaching Test (HJ 557-2010)

Figure 3 depicts the pH levels of the leachate and the leached Sb, as well as the immobilization efficiency of MBT and OPC samples, after 3 d, 7 d, 28 d, and 90 d of curing. In all samples, the leaching of Sb was higher than 5 μ/L (the Sb limit for the underground class III water according to GB/T 14848-2017 standard). The release of Sb increased with increasing curing time in the MBT and OPC specimens, and antimony shows high release after 28 days of curing. But the leaching of Sb in MBT sample was lower than that in OPC over the whole curing time range. Even if the leaching of Sb increased with curing time, a huge improvement between 3 d, 7 d, 28 d, and 90 d of immobilization could be seen. Besides, the immobilization efficiency for Sb after three and seven days of curing was up to 99%, and after 90 days, it was more than 90%. Based on the leaching results, it could be confirmed that MBT had the better curing performance of Sb. The higher UCS of MBT was attributed to the formation of more hydration products, which yielded a denser structure and thus reduced the release of Sb from tailings [24].

### 3.3. Microscopic Analysis

Figure 4 depicts the FE-SEM and EDS results for MBT and OPC after 28 days of aging. In both cases, the main hydration products were ettringite and C-S-H gel [25]. Compared with OPC, MBT exhibited a higher amount of C-S-H gel after 28 days of hydration, while OPC was rich in ettringite. In this respect, the above discussed high compressive strength of MBT (Figure 1) was attributed to the fact that ettringite acted as a skeleton and C-S-H gel served as a filler, making the structure of MBT denser. However, ettringite crystals in OPC were larger than in MBT. As a result, OPC had a less dense structure than MBT. Moreover, the EDS data (Figure 5) revealed that C-S-H gel was the main host mineral of Sb. Since C-S-H gel has a very high specific surface energy [22], its adsorption and physical encapsulation might be the main reason for why the fixing performance of antimony with MBT was better than that with OPC.

### 3.4. XRD Results

XRD patterns of two pastes after different periods of aging are shown in Figure 6. The main hydration products of pure MSB were ettringite and C-S-H gel, where the peak (2θ = 26.6°) intensity associated with ettringite increased over time. However, the signal from ettringite in MSB-Sb showed an opposite trend, gradually decreasing with aging, just as the peaks at 2θ close to 10° and 26.6°. In addition, the peak positions of ettringite and C-S-H gel were slightly offset in MSB-Sb after 28 days of curing. No antimony phase was found in MSB-Sb. This was because calcium antimony has a highly variable structure, both crystalline and amorphous. In addition, it might also be because of the low content of crystalline calcium antimony falling below the detection limit of XRD, which is generally 1–2 wt. % [26].

### 3.5. FT-IR Results

Figure 7 displays the FT-IR spectra of MSB-Sb and pure MSB pastes after 3 and 28 days of hydration. In both cases, the spectrograms were rather similar, presenting analogous absorption bands. The wide bands around 3624 cm^−1^ and 3427 cm^−1^ are associated with the vibration of OH-groups in water, whereas the band at 1656 cm^−1^ is a well-defined H-O-H deformation band from interlayer water [27]. The peak at 1114 cm^−1^ can be attributed to a [SO_4_] stretching mode (v3) [28]. In addition, the bands at 611 cm^−1^ and 667 cm^−1^ are related to the symmetric stretching and bending vibration of Al-OH species in the [Al(OH)_6_]^3−^ substructure of ettringite. The feature at 977 cm^−1^ is the antisymmetric Si-O(Al) stretching vibration which is characteristic of C-S-H gel. The bands at 513 cm^−1^ and 451 cm^−1^ are caused by the out-of-plane banding vibration and in-plane bending vibration of Si-O, respectively [14,29,30,31]. The only difference between MSB-Sb and MSB was that the absorption peaks related to C-S-H gel in MSB-Sb shifted to the wavelet numbers. This indicated that the incorporation of Sb into the cementing material had a significant effect on the hydration product C-S-H gel, which was manifested by a decrease in the degree of polymerization of C-S-H [17].

### 3.6. XPS Spectra

Table 3 displays the binding energies of main elements in the Sb cured sample and the blank group after 28 days of curing. As seen from the table, the binding energies of Ca, O, and Si atoms in the Sb cured sample slightly decreased with respect to those the reference groups, indicating that the incorporation of Sb exerted a great influence on silicate. Moreover, the covalent radius of Sb atom (1.39 Å) is larger than those of Al (1.21 Å) and Si (1.11 Å), making it difficult for Sb to undergo homogeneity substitution with Al and Si in C-S(A)-H gel [32]. In addition, Sb has a high electronegativity (1.9) [19]. If Sb enters the silicon (aluminum) oxygen tetrahedron, it will inevitably lead to a decrease in the outer electron cloud density of other surrounding ions and an increase in the binding energy of atoms [33]. However, according to Table 3, the binding energy of Al remained unchanged, while that of Si decreased during aging. Combined with the above SEM and IR results (Figure 4 and Figure 7), one may conclude that Sb was mainly adsorbed and wrapped in C-S (A)-H gel.

## 4. Discussions

Among the cementitious materials, the BFS with vitreous structure has strong chemical activity. The vitreous body of the slag is mainly composed of silicon–oxygen tetrahedrons, and some Al^3+^ replaces Si^4+^ to form alumino–oxytetrahedrons with higher activity than silicon-–oxygen tetrahedra, and a small amount of aluminosilicate crystallites with extremely low crystallinity [34].

Figure 8 displays the hydration reaction of MSB and it can be described as follows:(1)$$C_{2}{S+H}_{2}O\to C\text{-}S\text{-}H+\mathrm{Ca}{\left(\mathrm{OH}\right)}_{2}$$
(2)$$C_{3}{S+H}_{2}O\to C\text{-}S\text{-}H+\mathrm{Ca}{\left(\mathrm{OH}\right)}_{2}$$
(3)$$C_{3}{A+\mathrm{CaSO}}_{4}\cdot {2H}_{2}{O+H}_{2}O\to C_{6}{\mathrm{AS}}_{3}H_{32}$$
(4)$${\mathrm{Ca}}^{2+}{+\mathrm{Mg}}^{2+}+\mathrm{Sb}{\left(\mathrm{OH}\right)}_{6}^{-}{+H}_{2}O\to {\left(\mathrm{Ca},\mathrm{Mg}\right)}_{1.13}{\mathrm{Sb}}_{2}O_{6}{\mathrm{OH}}_{0.26}\cdot 0{.47H}_{2}O$$

In the early stage of hydration, the amorphous vitreous structure in the blast furnace slag dissociates and dissolves silicate, aluminate, and Ca^2+^, Al^3+^, Mg^2+^ ions, and reacts to form C-S(A)-H gel, such as (1) and (2). With the hydration and dissolution of the steel slag, the hydroxyl groups provide a strong alkaline environment for the system. The gel formed on the surface of the slag gradually separates, which makes the polymerization degree of the glass surface of the slag drop rapidly, and thus the remaining glass activity is reactivated, promoting the continuous hydration reaction of the slag While the slag hydration continuously consumes hydroxyl groups, the hydration of the steel slag is further promoted. Desulfurized gypsum provides a large amount of Ca^2+^ and SO_4_^2−^ ions for the system and enters the reactions with the gel to form ettringite (such as (3)), and the consumption of the gel promotes the hydration of the above-mentioned slag and steel slag [35]. The coordinated excitation of blast furnace slag, steel slag, and desulfurized gypsum is conducive to the mass production of ettringite and C-S-H gel. The C-S-H gel accumulates and encapsulates the ettringite minerals to form a tight and stable structure [28]. Moreover, the large specific surface area of C-S-H gel has great potential to adsorb heavy metals. Although the surface is negatively charged, due to the charge balance, the surface charge is reversed after the adsorption of calcium ions, allowing antimony anions to be adsorbed [36,37,38,39]. In addition, antimony precipitates can be tightly encapsulated and sealed by C-S-H gel. Ca[Sb(OH)_6_]_2_ precipitates tend to be in equilibrium at the lower pH, while precipitates at higher pH scales are more likely to be romeites (Ca_1+x_Sb_2_O_6_OH_2−2x_) [40], such as (4) [13]. It is different from the incorporation in the ettringite for Sb(OH)_6_^−^, which has a large octahedral structure. Therefore, the valid immobilization mechanism for Sb(OH)_6_^−^ is to be adsorbed on the ettringite surface rather than incorporated into the structure [31,40]. In this study, the adsorption of Sb by ettringite was not proved and needs to be further verified in the synthesis of single mineral ettringite.

According to the current knowledge, in Hebei and Guangxi provinces, the purchase price of raw materials SS, BFS, and FGDG is about 10 yuan/t, 120 yuan/t, and 15 yuan/t respectively, while SS is no charged in many areas. The original price of MSB material is 10 × 0.3 + 120 × 0.6 + 15 × 0.1 = 76.5 yuan/t, plus 130 yuan of processing fee and 20% profit of enterprises, and the price of MSB is 247.8 yuan. This is much lower than 450 yuan/t of cement. In addition, it also avoids CO_2_ emissions from calcination of limestone during cement production.

## 5. Conclusions

The solidified bodies of metallurgical slag and ordinary Portland cement (OPC)-solidified antimony were compared via compressive strength measurements and horizontal oscillation method, and pure slurry was prepared to elucidate the mineralogical solidification mechanism of metallurgical slag to antimony. The following conclusions were drawn:Green mining fill samples (MBT) exhibited higher strength and a more pronounced antimony fixation effect than those of OPC.No newly formed antimony-containing mineral phase was detected in the metallurgical slag, but it was mainly surrounded by the adsorbed gel.Even though the heavy metal antimony was found to be able to reduce the degree of polymerization in the gel, its influence on ettringite and its relationship are not yet proven.

## Figures and Tables

**Figure 1 materials-15-01780-f001:**
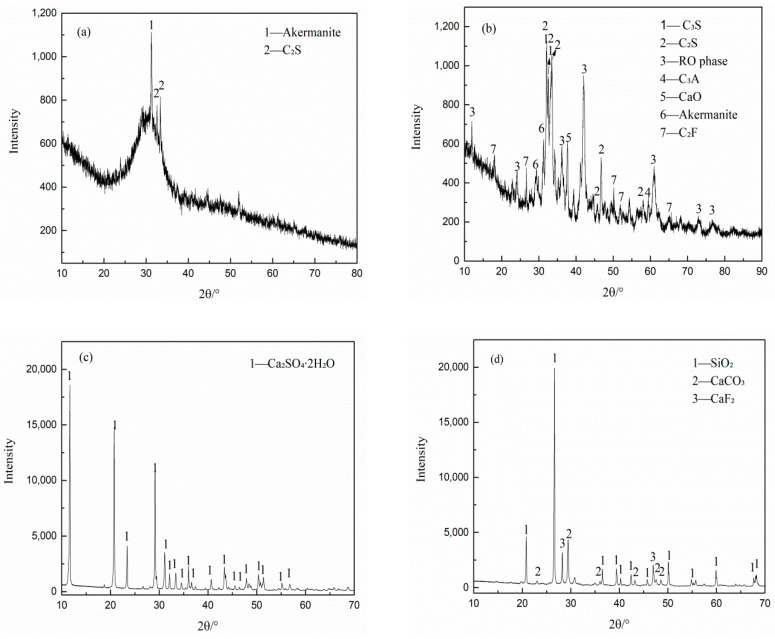
X-ray diffraction (XRD) patterns of raw materials: (**a**) blast furnace slag (BFS); (**b**) steel slag (SS); (**c**) flue gas desulfurized gypsum (FGDG); (**d**) high Sb-containing mine tailings (Sb-MT).

**Figure 2 materials-15-01780-f002:**
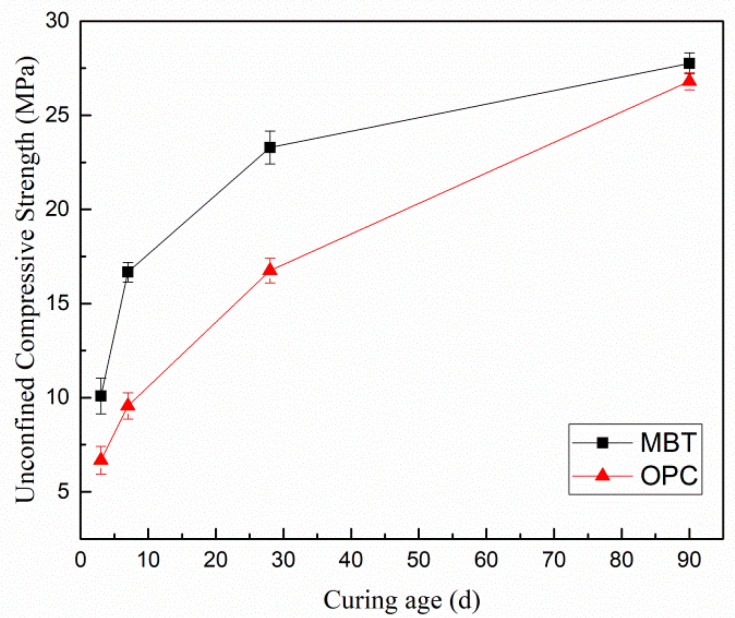
Unconfined compressive strength (UCS) results of MBT and OPC at different curing times.

**Figure 3 materials-15-01780-f003:**
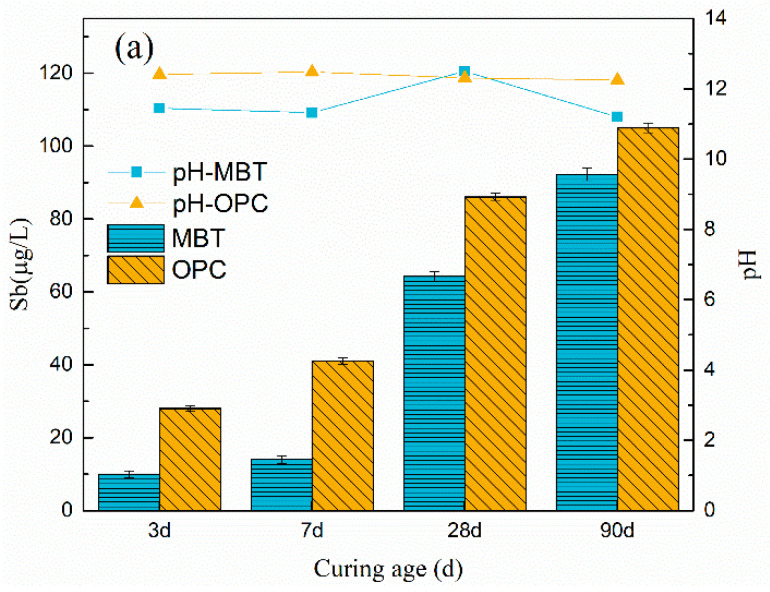
(**a**) Evolution of pH and Sb release for each MBT and ordinary Portland cement (OPC) sample; (**b**) immobilization efficiency of Sb after 3 d, 7 d, 28 d, and 90 d of curing during horizontal vibration leaching method.

**Figure 4 materials-15-01780-f004:**
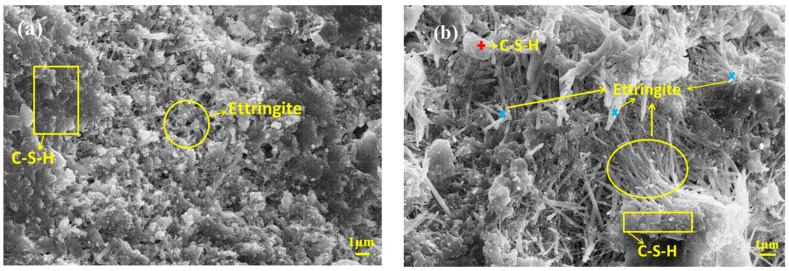
Scanning electron microscope (SEM) images of MBT with OPC at curing age of 28 days: (**a**) MBT (**b**) OPC.

**Figure 5 materials-15-01780-f005:**
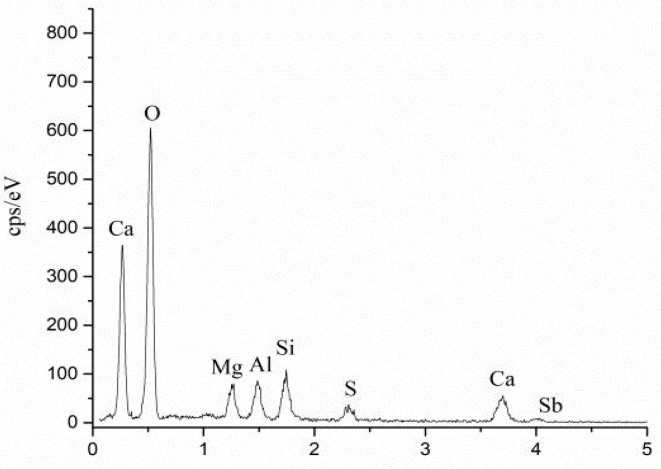
Area of MBT-C-S-H at curing age of 28 days.

**Figure 6 materials-15-01780-f006:**
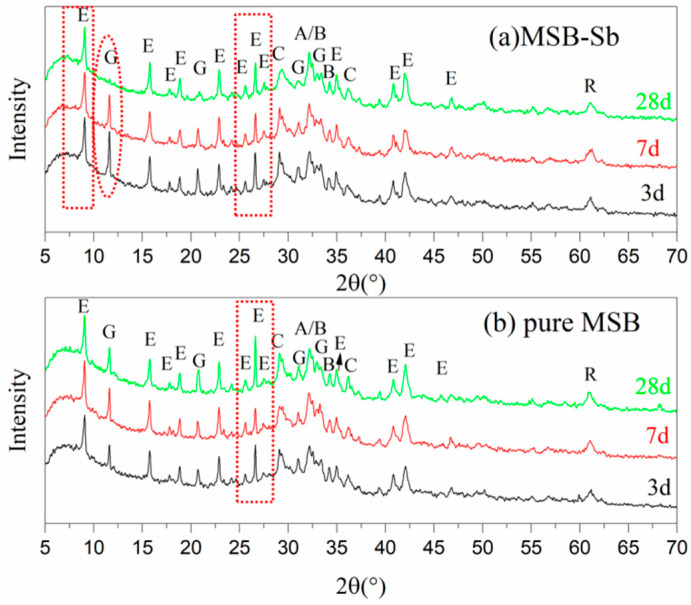
XRD patterns of pastes after 3 d, 7 d, and 28 d of aging. (**a**)MSB-Sb; (**b**) pure MSB. Here, phase designations are as follows: A: Alite (Ca_3_SiO_5_, C_3_S); B: Belite (Ca_2_SiO_2_, C_2_S); C: C-S-H gel; E: Ettringite (Ca_6_Al_2_(SO_4_)_3_(OH)_12_∙26H_2_O); G: Gypsum (CaSO_4_∙2H_2_O); R: RO phase.

**Figure 7 materials-15-01780-f007:**
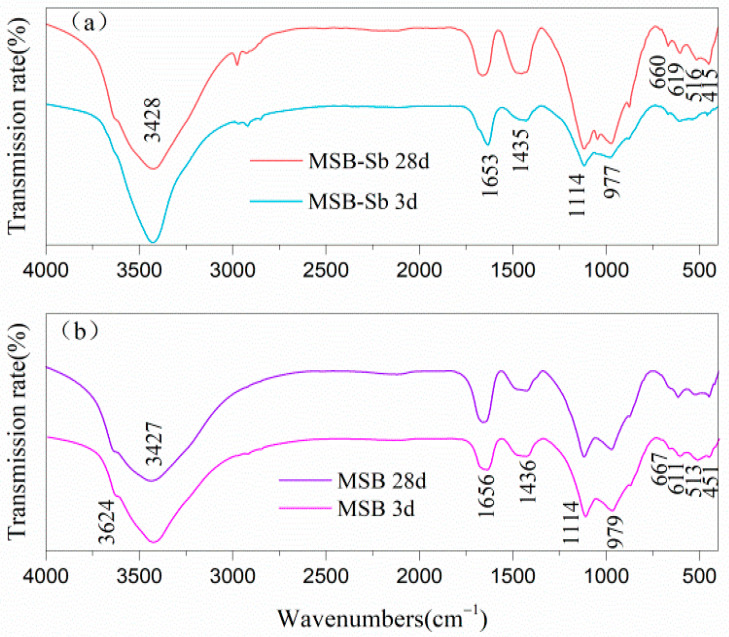
FT-IR spectra of pastes after 3 and 28 days of curing: (**a**) MSB with Sb; (**b**) pure MSB.

**Figure 8 materials-15-01780-f008:**
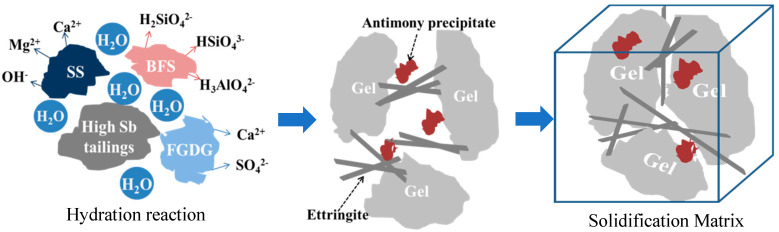
Schematic diagram of curing mechanism of Sb in metallurgical-slag-based materials.

**Table 1 materials-15-01780-t001:** Raw material chemical components, physical properties, concentrations, and leaching concentration of Sb.

Materials		Sb-MT	BFS	SS	FGDG
Chemical composition	MgO	0.99	8.94	6.00	1.04
Oxide (wt. %)	Al_2_O_3_	4.45	15.43	6.24	0.78
	SiO_2_	40.34	24.76	18.16	2.03
	SO_3_	17.31	0.83	0.29	44.97
	CaO	15.15	46.16	42.58	45.31
	Fe_2_O_3_	17.52	2.52	17.66	0.48
Blaine fineness (m^2^/Kg)		-	400	400	360
pH		7.35	11.92	12.28	7.85
Leaching Sb concentration (μg/L)		524	ND	ND	ND

**Table 2 materials-15-01780-t002:** Proportions of total solid waste S/S materials (MSB) components, including equivalent binder-tailings ratio and solid concentration of green mining fill (MBT) samples.

Notation	MSB (Mass Fraction/wt. %)			Binders/Tailings (*w*/*w*)	Solid Concentration ^a^ (wt. %)
	BFS	SS	FGDG		
MBT	60	30	10	¼	86
OPC ^b^	100(OPC)				

^a^ Solid concentration = (binders + tailings)/(binders + tailings + water). ^b^ OPC used as a cementitious material for the control group.

**Table 3 materials-15-01780-t003:** Changes in binding energies of main elements in Sb cured sample and blank group after 28 days of curing.

Elements	Ca	Al	Si	O	S
Blank group	347.30	74.26	102.10	531.67	168.99
Sb cured sample	347.11	74.26	102.00	531.54	169.21
Changes	−0.19	0	−0.10	−0.13	+0.22

## Data Availability

Not applicable.

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
