# Peer review of "Study on Solidification and Stabilization of Antimony-Containing Tailings with Metallurgical Slag-Based Binders"

_materials, 2022, doi:10.3390/ma15051780_

Round 1

Reviewer 1 Report

The paper presents a study on the solidification and stabilization of antimony-containing tailings with metallurgical slag-based binders.
In general the text is easy to read and understandable. Unfortunately it was not possible to conclude whether antimony has entered the crystal structure of ettringite.
As corrections I suggest:
In these minor corrections I will only inform the line. These are predominantly typos and can be identified in the line:
Lines: 35, 36, 57, 198, 262, 318, 319, 341, 347, 369, 372, 373 and 376.
Specific comments:
In Figure 1 letter "a" the diffractogram has several non-indexed peaks and in letter "c" the peaks assigned to C2S are not clear.
In the results item "3.4 XRD Results" it is also not clear the observation of the sentence from line 198 to 201 about variations in the intensities of the ettringite peak. It is not convincing.
In the References I suggest a major revision, for example, in how the authors' names are being presented.
Otherwise the article can be published, it is organized, easy to understand what the authors were getting at, just a pity that they have not yet been able to prove where antimony really is chemically inserted.

Author Response

Dear reviewer:

I am very grateful to your comments for the manuscript. According with your advice, we amended the relevant part in manuscript. Some of your questions were answered below:

  1. In Figure 1 letter "a" the diffractogram has several non-indexed peaks and in letter "c" the peaks assigned to C2S are not clear.

The Figure 1 has been modified.

Figure 1. XRD patterns of the raw materials: (a) BFS; (b) SS; (c) FGDG; (d) Sb-MT.

  1. In the results item "3.4 XRD Results" it is also not clear the observation of the sentence from line 198 to 201 about variations in the intensities of the ettringite peak. It is not convincing.

XRD patterns of two pastes after different periods of aging are shown in Fig. 6. The main hydration products of pure MSB were ettringite and C-S-H gel, where the peak (2θ=26.6°) intensity associated with ettringite increased over time. However, the signal from ettringite in MSB-Sb showed an opposite trend, gradually decreasing with aging, just as the peaks at 2θ close to 10° and 26.6°.

Figure.6. XRD patterns of pastes after 3 d, 7 d, and 28 d of aging. Here, the phase designations are as follows: A: Alite (Ca3SiO5, C3S); B: Belite (Ca2SiO2, C2S); C: C-S-H gel; E: Ettringite (Ca6Al2(SO4)3(OH)12∙26H2O); G:Gypsum (CaSO4∙2H2O); R: RO phase.

  1. In the References I suggest a major revision, for example, in how the authors' names are being presented.

The format of references has been unified.

[1]            Nishad, P.A.; Bhaskarapillai N. Antimony, a pollutant of emerging concern: a review on industrial sources and remediation technologies[J]. Chemosphere, 2021,277: 130252.

( https://doi.org/10.1016/j.chemosphere.2021.130252)

[2]            Ashley, P.M.; Craw, D.; Graham, B.P.; Chappell, D.A. Environmental mobility of antimony around mesothermal stibnite deposits, New South Wales, Australia and southern New Zealand[J]. Journal of Geochemical Exploration, 2003.77(1):1-14

(https://doi.org/10.1016/S0375-6742(02)00251-0)

[3]            Zhang, Y.; Ding, C.X.; Gong, D.X.; Deng, Y.C.; Huang, Y.; Zheng, J.F.; Xiong, S.;Tang, R.D.; Wang, Y.C.; Su, L. A review of the environmental chemical behavior, detection and treatment of antimony[J]. Environment Technology & Innovation. 2021,24: 102026.( https://doi.org/10.1016/j.eti.2021.102026)

[4]            Corneils, G.; Gerven, T.V.; Vandecasteele, C. Antimony leaching from MSWI bottom ash: Modelling of the effect of pH and carbonation[J]. Waste Management, 2012,32(2): 278-286.( https://doi.org/10.1016/j.wasman.2011.09.018)

[5]            Guo, J.L.; Yin, Z.P.; Zhong, W.; Jing C.Y. Immobilization and transformation of co-existing arsenic and antimony in highly contaminated sediment by nano zero-valent iron[J]. Journal of Environmental Sciences, 2022,112: 152-160.( https://doi.org/10.1016/j.jes.2021.05.007)

[6]            Xiao, B.L.; Wen, Z.J.; Miao S.J. Utilization of steel slag for cemented tailings backfill: Hydration, strength, pore structure, and cost analysis[J]. Case Studies in Construction Materials, 2021,15,e00621.( https://doi.org/10.1016/j.cscm.2021.e00621)

[7]            Almas, A.R.; Pironin, E.; Okkenhaug, G. The partitioning of Sb in contaminated soils after being immobilization by Fe-based amendments is more dynamic compared to Pb[J]. Applied Geochemistry, 2019,108: 104378.( https://doi.org/10.1016/j.apgeochem.2019.104378)

[8]            Andrew, R.M. Global CO2 Emissions from Cement Production[J]. Earth System Science Data Discussions, 2018: 1-61.( https://doi.org/10.5194/essd-2017-77)

[9]      Zhang, C.Y.; Han, R.; Yu, B.Y.; Wei, Y.M. Accounting process-related CO2 emissions from global cement production under Shared Socioeconomic Pathways[J].Journal of Cleaner Production, 2018,184:451-465.( https://doi.org/10.1016/j.jclepro.2018.02.284)

[10]          Li, J. S.; Chen, L.; Zhan, B.J.; Wang, L.; Poon, C.S.; Daniel C.W.Tsang. Sustainable stabilization/solidification of arsenic-containing soil by blast slag and cement blends[J]. Chemosphere, 2021,271(10): 129868.( https://doi.org/10.1016/j.chemosphere.2021.129868)

[11]          Wang, L.; Chen, L.; Cho D.W.; Daniel C.W.Tsang; Yang, J.; Hou, D.Y.; Baek, K.; Kua, H.W.; Poon, C.S. Novel synergy of Si-rich minerals and reactive MgO for stabilisation/solidification of contaminated sediment[J]. Journal of Hazardous Materials, 2019,365(5): 695-706.( https://doi.org/10.1016/j.jhazmat.2018.11.067)

[12]          Salihoglu, G. Immobilization of antimony waste slag by applying geopolymerization and stabilization/solidification technologies[J]. Journal of the Air & Waste Management Association, 2014,64(11):1288-1298.( https://doi.org/10.1080/10962247.2014.943352)

[13]          Cornelis, G.; Etschmann, B.; Gerven, T.V.; Vandecasteenle, C. Mechanisms and modelling of antimonate leaching in hydrated cement paste suspensions[J]. Cement and Concrete Research, 2012,42(10): 1307-1316.( https://doi.org/10.1016/j.cemconres.2012.06.004)

[14]          Gao, W.; Li, Z.F.; Zhang, S.Q.; Zhang Y.Y.; Fu, P.F.; Yang H.F.; Ni, W. Enhancing Arsenic Solidification/Stabilisation Efficiency of Metallurgical Slag-Based Green Mining Fill and Its Structure Analysis[J]. metals, 2021,11: 1389.( https:// doi.org/10.3390/met11091389)

[15]          Gao, W.; Li, Z.F.; Zhang, S.Q.; Zhang Y.Y.; Teng, G.X.; Li, X.Q.; Ni, W.Solidification/Stabilization of Arsenic-Containing Tailings by Steel Slag-Based Binders with High Efficiency and Low Carbon Footprint[J]. materials, 2021,14: 5864.( https:// doi.org/10.3390/ma14195864)

[16]          Rasaki, S.A.; Zhang, B.X.; Guarecuco, R.; Thomas, T.J.; Yang, M.H. Geopolymer for use in heavy metals adsorption, and advanced oxidative processes: A critical review[J]. Journal of Cleaner Production, 2019,213: 42-58.( https://doi.org/10.1016/j.jclepro.2018.12.145)

[17]          Li, J.; Zhang, S.Q.; Wang, Q.; Ni, W.; Li, K.Q.; Fu, P.F.; Hu, W.T.; Li Z.F. Feasibility of using fly ash–slag-based binder for mine backfilling and its_associated leaching risks[J]. Journal of Hazardous Materials, 2020,400: 123191.

( https://doi.org/10.1016/j.jhazmat.2020.123191)

[18]          Li, Y.Y.; Ni, W.; Gao, W.; Zhang, Y.Y.; Yan, Q.H.; Zhang, S.Q. Corrosion evaluation of steel slag based on a leaching solution test[J]. Energy Sources, Part A: Recovery, Utilization, and Environmental Effects, 2019,41(7): 790-801.

( https://doi.org/10.1080/15567036.2018.1520359)

[19]          Cornelis, G.; Gerven, T.V.; Vandecasteele, C. Antimony leaching from MSWI bottom ash Modelling of the effect ofpH and carbonation[J]. Waste management, 2012,32(2): 278-286.( https://doi.org/10.1016/j.wasman.2011.09.018)

[20]          Cornelis, G.; Johnson, C.A.; Gerven, T.V.; Vandecasteele, C. Leaching mechanisms of oxyanionic metalloid and metal species in alkaline solid wastes[J]. Applied Geochemistry, 2008,23(5): 955-976.( https://doi.org/10.1016/j.apgeochem.2008.02.001)

[21]          Gao, W.; Ni, W.; Zhang, Y.Y.; Li, Y.Y.; Shi, T.Y.; Li, Z.F. Investigation into the semi-dynamic leaching characteristics of arsenic and antimony from solidified/stabilized tailings using metallurgical slag-based binders[J]. Journal of Hazardous Materials, 2020,381: 120992.( https://doi.org/10.1016/j.jhazmat.2019.120992)

[22]          Zhang, Y.Y.; Zhang, S.Q.; Ni, W.; Yan, Q.H.; Gao, W.; Li, Y.Y. Immobilisation of high-arsenic-containing tailings by using metallurgical slag-cementing materials[J]. Chemosphere, 2019,223:117-123.( https://doi.org/10.1016/j.chemosphere.2019.02.030)

[23]          Zhang, Y.Y.; Gao, W.; Ni, W.; Zhang, S.Q.; Li, Y.Y.; Wang K.; Huang, X.H.; Fu, P.F.; Hu W.T. Influence of calcium hydroxide addition on arsenic leaching and solidification/stabilisation behaviour of metallurgical-slag-based green mining fill[J]. Journal of Hazardous Materials, 2020,390: 122161.

( https://doi.org/10.1016/j.jhazmat.2020.122161)

[24]          Yan, Q.H.; Ni, W.; Gao, W.; Li, Y.Y.; Zhang, Y.Y. Mechanism for solidification of arsenic with blast furnace slag-steel slag based cementitious materials[J]. Journal of Central South University(Science and Technology), 2019,50(07): 1544-1550.

( https://doi.org/10.11817/j.issn.1672-7207.2019.07.006)

[25]         Li, Y.C.; Wu, J.X.; Hu, W.; Ren, B.Z.; Hursthouse, A.S. A mechanistic analysis of the influence of iron-oxidizing bacteria on antimony (V) removal from water by microscale zero-valent iron[J]. Journal of Chemical Technology & Biotechnology, 2018,93.(https://doi.org/10.1002/jctb.5606)

[26]          Yi, H.; Xu, G.P.; Cheng, H.G.; Wang, J.S.; Wan, Y.F.; Chen, H. An Overview of Utilization of Steel Slag[J]. Procedia Environmental Sciences, 2012,16: 791-801.( https://doi.org/10.1016/j.proenv.2012.10.108)

[27]          Li, Y.;  Qiao, C.y.; Ni W. Green concrete with ground granulated blast-furnace slag activated by desulfurization gypsum and electric arc furnace reducing slag[J]. Journal of Cleaner Production, 2020,269: 122212.( https://doi.org/10.1016/j.jclepro.2020.122212)

[28]          Liu, Y.; Ni, W.; Huang, X.Y.; Ma, X.M.; Li, D.Z. Characteristics of hydration and hardening of red mud of bayer process in carbide slag-flue gas desulfurization gypsum system[J]. Materials Reports, 2016,30(14): 120-124.

(https://doi.org/10.11896/j.issn.1005-023X.2016.14.027)

[29]          Ni, W.; Li, Y.; Xu, D.; Xu, C.W.; Jiang, Y.Q.; Gao, G.J. Hydration mechanism of blast furnace slag-reduction slag based solid waste cementing materials[J]. Journal of Central South University(Science and Technology), 2019,50(10): 2342-2351.

( https://doi.org/10.11817/j.issn.1672-7207.2019.10.002)

[30]          Eskander, S.B.; Bayoumi, T.A.; Tawfik, M.E. Immobilization of Borate Waste Simulate in Cement-Water Extended Polyester Composite Based on Poly(Ethylene Terephthalate) Waste [J]. Polymer-Plastics Technology and Engineering, 2006,45(8): 939-945.

( https://doi.org/10.1080/03602550600723415)

[31]          Zhao, Y,L.; Wu, P.Q.; Qiu J.P.; Guo, Z.B.; Tian, Y,S.; Sun, X,G,; Gu, X.W. Recycling hazardous steel slag after thermal treatment to produce a binder for cemented paste backfill[J]. Powder Technology, 2022,395: 652-662.

( https://doi.org/10.1016/j.powtec.2021.10.008)

[32]          Ines, G.L.; Fernandez-Jimenez A.; Blanco, M.T.; Palomo, A. FTIR study of the sol–gel synthesis of cementitious gels: C–S–H and N–A–S–H[J]. Journal of Sol-Gel Science and Technology, 2008,45(1): 63-72.( https://doi.org/10.1007/s10971-007-1643-6)

[33]          Noam, A. Covalent radii from ionization energies of isoelectronic series[J]. Chemical Physics Letters, 2014,595-596: 214-219.( https://doi.org/10.1016/j.cplett.2014.01.037)

[34]     Cornelis, G.; Johnson, C.A.; Gerven, T.V.; Vandecasteenle, C. Leaching mechanisms of oxyanionic metalloid and metal species in alkaline solid wastes: A review[J]. Applied Geochemistry, 2008,23(5): 955-976.( https://doi.org/10.1016/j.apgeochem.2008.02.001)

[35]          Santamaria, L.; Vicente, M.A.; Korili, S.A.; Gil, A. Effect of the preparation method and metal content on the synthesis of metal modified titanium oxide used for the removal of salicylic acid under UV light[J]. Environmental Technology, 2018,41: 2073-2084.( https://doi.org/10.1080/09593330.2018.1555285)

[36]          Li, Y.; Wu, B.H.; Ni, W.; Li, X.M. Synergies in Early Hydration Reaction of Slag-Steel Slag-Gypsum System[J]. Journal of Northeastern University( Natural Science), 2020,41(4): 581-586.( https://doi.org/10.12068/j.issn.1005-3026.2020.04.022)

[37]          Avalos, N.M.; Varga, T.; Mergelsberg, S.T.; Silverstein, J.A.; Saslow, A.A. Behavior of iodate substituted ettringite during aqueous leaching[J]. Applied Geochemistry, 2021,125: 104863.( https://doi.org/10.1016/j.apgeochem.2020.104863)

[38]          Pointeau, I.; Reiller, P.; Mace, N.; Landesman, C.; Coreau. N. Measurement and modeling of the surface potential evolution of hydrated cement pastes as a function of degradation[J]. Journal of Colloid & Interface Science, 2006,300(1): 33-44.( https://doi.org/10.1016/j.jcis.2006.03.018)

[39]          Zou, C.; Long, G.C.; Zheng, X.H.; Ma, C.; Xie, Y.J.; Sun, Z.P. Water evolution and hydration kinetics of cement paste under steam-curing condition based on low-field NMR method[J]. Construction and Building Materials, 2020,271: 121583.( https://doi.org/10.1016/j.conbuildmat.2020.121583)

[40]          Elakneswaran, Y; Nawa, T.; Kurumisawa, K. Electrokinetic potential of hydrated cement in relation to adsorption of chlorides[J]. Cement & Concrete Research, 2009,39(4): 340-344.( https://doi.org/10.1016/j.cemconres.2009.01.006)

[41]          Ochs, M.; Pointeau, I.; Giffaut, E. Caesium sorption by hydrated cement as a function of degradation state: Experiments and modelling[J]. Waste Management, 2006,26(7): 725-732.( https://doi.org/10.1016/j.wasman.2006.01.033)

[42]          Cornelis, G.; Gerven, T.V.; Snellings, R.; Verbinnen, B.; Elsen, J.; Vandecasteele, C. Stability of pyrochlores in alkaline matrices Solubility of calcium antimonate[J]. Applied Geochemistry.2011,26(5): 809-817.( https://doi.org/10.1016/j.apgeochem.2011.02.002)

Reviewer 2 Report

Dear authors! 
Your manuscript shown the solidifying mechanism of tailings containing high concentration of Sb with metallurgical slag-based binders. The experimental work was performed thoroughly, but some results and conclusion is not clearly. 
Some comments placed below:
You should shortly describe your previous articles results and indicate the difference with this manuscript.
Line 37-38: Please, add the reference.
Line 73-76: You should consider articles about the solidification of antimony tailings by cement- metallurgical waste binders, such as  https://doi.org/10.1080/10962247.2014.943352, https://doi.org/10.1080/10643389.2021.1944588
Line 100: I see Al and Mg in the chemical composition of Sb-MT, but you did not indicate minerals with these elements XRD pattern. There are some not described peaks in the XRD pattern of Sb-MT. 
Line 104: Why are figure one from this manuscript and figure one from your previous article (https://doi.org/10.3390/ma14195864) almost identical? The chemical composition of materials is significantly different in these works. You should explain it.
Line 179 (Figure 3 (b)): The difference between the immobilization efficiency of OPC and MBT is insignificant.

Author Response

Dear reviewer:

I am very grateful to your comments for the manuscript. According with your advice, we amended the relevant part in manuscript. Some of your questions were answered below:

  1. Line 37-38: Please, add the reference.

This paragraph is revised as follows:

In 2018, China produced 12.11 billion tons of tailings, accounting for 35.11% of the total production of bulk industrial solid waste. The tailings after mineral processing are usually discarded as solid waste in the tailings ponds without any treatment, carrying potential heavy metal pollution risks [1]. Antimony is a toxic metalloid in the group VA of the fifth period of the periodic table. Antimony used in the lead-zinc tailings is mainly stibnite (Sb2S3) [2], and many studies have reported that the migration and transformation of antimony is mainly Sb(OH)6- [3, 4]. In this respect, a series of reactions such as oxidation and leaching occurs under the combined action of oxygen, water and microorganisms, making Sb2S3 soluble SbO3- and hydrolyze to Sb(OH)6-, and then attached to the surface of various solid particles in the tailings in the form of adsorption [3, 5]. This part of antimony usually has high activity, and without proper pre-treatment managed, it can cause serious or even catastrophic consequences.

  1. Line 73-76: You should consider articles about the solidification of antimony tailings by cement- metallurgical waste binders, such as  https://doi.org/10.1080/10962247.2014.943352, https://doi.org/10.1080/10643389.2021.1944588

This paragraph is revised as follows:

Ordinary Portland cement (OPC) is traditionally used for landfill solidification/stabilization (S/S), and has long-term stability [8-11].For example, there are research reports used Portland cement, fly ash, clay, gypsum and blast furnace slag as a solidification/stabilization materials to limit the leaching potential of antimony [12]. According to their results, the leaching rate of antimony-containing waste residue was low and fulfilled the landfill standard and gypsum was successful in immobilizing the antimony. Cornelis investigated the leaching characteristic of antimony by mixing Portland cement with KSb(OH)6 [13]. They founded that various hydration minerals of cement had limited effects on antimony, such as AFm minerals and C-S-H gel. Unfortunately they didn’t do a mineralogical analysis to verify the solidifying mechanism of antimony.

  1. Line 100: I see Al and Mg in the chemical composition of Sb-MT, but you did not indicate minerals with these elements XRD pattern. There are some not described peaks in the XRD pattern of Sb-MT. 

XRD results show that there is no phase containing Al and Mg, which is due to:

  1. The chemical composition analysis of tailings contains Al and Mg, but the component content is very low, which may be lower than the detection limit of XRD. 2. The intensity of quartz peak is very strong, which will interfere with the relatively weak peak. 3. In addition, Al and Mg may not exist in the form of crystal.
  2. Line 104: Why are figure one from this manuscript and figure one from your previous article (https://doi.org/10.3390/ma14195864) almost identical? The chemical composition of materials is significantly different in these works. You should explain it.

We use the same raw materials, but the difference is that the batches of raw materials are different. I had redrawn the figure one.

Figure 1. XRD patterns of the raw materials: (a) BFS; (b) SS; (c) FGDG; (d) Sb-MT.

  1. Line 179 (Figure 3 (b)): The difference between the immobilization efficiency of OPC and MBT is insignificant.

Figure 3(b) immobilization efficiency of OPC and MBT was revised. Although the difference is not obvious in the later curing time, the immobilization efficiency of the MBT is significantly better than that of OPC.

Figure 3. (a) Evolution of pH and Sb release for each MBT and OPC sample; (b) immobilization efficiency of Sb after 3 d, 7 d, 28 d, and 90 d of curing during the horizontal vibration leaching method

Reviewer 3 Report

In the present study, the solidified bodies of metallurgical slag and OPC solidified antimony were compared via compressive strength measurements and horizontal oscillation method, and pure slurry was prepared to elucidate the mineralogical solidification mechanism of metallurgical slag to antimony. The work described in the manuscript of sufficient novelty, quality, and potential significance to warrant publication, the provided methods sufficient for an interested reader to reproduce the results, and the conclusions adequately supported by the data presented. So, the publication is recommended after revision:

Comments
Provide the numerical data in the significant findings to highlight the efficiency of the new method.

  1. Cost of such process needs to be reported and compared with previous related work.
  2. More profound discussions and comparisons with other published works are welcomed.
  3. procedures followed in the experimental section must be supported by references.
  4. The Manuscript needs thorough revision to improve the text quality and readability of the work.
  5. Please check the grammar, uniformity in reference format and spell-check is necessary throughout the manuscript.

Author Response

Dear reviewer:

I am very grateful to your comments for the manuscript. According with your advice, we amended the relevant part in manuscript. Some of your questions were answered below:

  1. Cost of such process needs to be reported and compared with previous related work.

According to the current knowledge, in Hebei and Guangxi provinces, the purchase price of raw materials SS, BFS, FGDG is about 10 yuan/t, 120 yuan/t and 15 yuan/t respectively, while SS is no charged in many areas. The original price of MSB material is 10*0.3+120*0.6+15*0.1=76.5 yuan/t, plus 130 yuan of processing fee and 20% profit of enterprises, and the price of MSB is 247.8 yuan. This is much lower than 450 yuan/t of cement. In addition, it also avoids CO2 emissions from calcination of limestone during cement production.

  1. More profound discussions and comparisons with other published works are welcomed.

For example, there are research reports used Portland cement, fly ash, clay, gypsum and blast furnace slag as a solidification/stabilization materials to limit the leaching potential of antimony[12]. According to their results, the leaching rate of antimony-containing waste residue was low and fulfilled the landfill standard and gypsum was successful in immobilizing the antimony. Cornelis investigated the leaching characteristic of antimony by mixing Portland cement with KSb(OH)6[13]. They founded that various hydration minerals of cement had limited effects on antimony, such as AFm minerals and C-S-H gel. Unfortunately they didn’t do a mineralogical analysis to verify the solidifying mechanism of antimony. However, the production of cement is accompanied by the large CO2 emission, yielding the high cost and limiting the use of the final product[9]. To reduce carbon emissions and to make it possible to absorb a large amount of metallurgical solid waste, blast furnace slag (BFS), steel slag (SS), and flue gas desulfurized gypsum (FGDG) were proposed as cementing binders[14-16]. In the hydration process, BFS provides potentially active silicon-oxygen and aluminum-oxygen tetraheda, SS is a source of alkaline substances, and FGDG ensure sulfate ions[17, 18]. This enables one to form ettringite with low solubility and amorphous C-S-H gels, which would enhance the physical and mechanical properties of the cured material to stabilize Sb[13]. At present, most of researches aim at evaluating the environmental friendliness of S/S materials and the leaching characteristics of heavy metals using different leaching methods [19, 20].

  1. procedures followed in the experimental section must be supported by references.

References have been added to the procedures.

The MBT pastes were poured into 50×50×50 mm3 steel molds and then cured in a curing chamber (at the relative humidity of approximately 90% and 40℃, mimicking the conditions of underground mine storage in Guangxi) [23]. The unconfined compressive strength (UCS) of the specimens was assessed as according to GB/T17671-1999 standard at curing times of 3 d, 7 d, 28 d, and 90 d. Toxicity characteristic leaching tests were conducted via the horizontal vibration method using MBT and OPC at curing times of 3 d, 7 d, 28 d, and 90 d [22].

  1. The Manuscript needs thorough revision to improve the text quality and readability of the work.

We worked hard to revise the language of the manuscript.

  1. Please check the grammar, uniformity in reference format and spell-check is necessary throughout the manuscript.

The format of references has been unified.

[1]            Nishad, P.A.; Bhaskarapillai N. Antimony, a pollutant of emerging concern: a review on industrial sources and remediation technologies[J]. Chemosphere, 2021,277: 130252.

( https://doi.org/10.1016/j.chemosphere.2021.130252)

[2]            Ashley, P.M.; Craw, D.; Graham, B.P.; Chappell, D.A. Environmental mobility of antimony around mesothermal stibnite deposits, New South Wales, Australia and southern New Zealand[J]. Journal of Geochemical Exploration, 2003.77(1):1-14

(https://doi.org/10.1016/S0375-6742(02)00251-0)

[3]            Zhang, Y.; Ding, C.X.; Gong, D.X.; Deng, Y.C.; Huang, Y.; Zheng, J.F.; Xiong, S.;Tang, R.D.; Wang, Y.C.; Su, L. A review of the environmental chemical behavior, detection and treatment of antimony[J]. Environment Technology & Innovation. 2021,24: 102026.( https://doi.org/10.1016/j.eti.2021.102026)

[4]            Corneils, G.; Gerven, T.V.; Vandecasteele, C. Antimony leaching from MSWI bottom ash: Modelling of the effect of pH and carbonation[J]. Waste Management, 2012,32(2): 278-286.( https://doi.org/10.1016/j.wasman.2011.09.018)

[5]            Guo, J.L.; Yin, Z.P.; Zhong, W.; Jing C.Y. Immobilization and transformation of co-existing arsenic and antimony in highly contaminated sediment by nano zero-valent iron[J]. Journal of Environmental Sciences, 2022,112: 152-160.( https://doi.org/10.1016/j.jes.2021.05.007)

[6]            Xiao, B.L.; Wen, Z.J.; Miao S.J. Utilization of steel slag for cemented tailings backfill: Hydration, strength, pore structure, and cost analysis[J]. Case Studies in Construction Materials, 2021,15,e00621.( https://doi.org/10.1016/j.cscm.2021.e00621)

[7]            Almas, A.R.; Pironin, E.; Okkenhaug, G. The partitioning of Sb in contaminated soils after being immobilization by Fe-based amendments is more dynamic compared to Pb[J]. Applied Geochemistry, 2019,108: 104378.( https://doi.org/10.1016/j.apgeochem.2019.104378)

[8]            Andrew, R.M. Global CO2 Emissions from Cement Production[J]. Earth System Science Data Discussions, 2018: 1-61.( https://doi.org/10.5194/essd-2017-77)

[9]      Zhang, C.Y.; Han, R.; Yu, B.Y.; Wei, Y.M. Accounting process-related CO2 emissions from global cement production under Shared Socioeconomic Pathways[J].Journal of Cleaner Production, 2018,184:451-465.( https://doi.org/10.1016/j.jclepro.2018.02.284)

[10]          Li, J. S.; Chen, L.; Zhan, B.J.; Wang, L.; Poon, C.S.; Daniel C.W.Tsang. Sustainable stabilization/solidification of arsenic-containing soil by blast slag and cement blends[J]. Chemosphere, 2021,271(10): 129868.( https://doi.org/10.1016/j.chemosphere.2021.129868)

[11]          Wang, L.; Chen, L.; Cho D.W.; Daniel C.W.Tsang; Yang, J.; Hou, D.Y.; Baek, K.; Kua, H.W.; Poon, C.S. Novel synergy of Si-rich minerals and reactive MgO for stabilisation/solidification of contaminated sediment[J]. Journal of Hazardous Materials, 2019,365(5): 695-706.( https://doi.org/10.1016/j.jhazmat.2018.11.067)

[12]          Salihoglu, G. Immobilization of antimony waste slag by applying geopolymerization and stabilization/solidification technologies[J]. Journal of the Air & Waste Management Association, 2014,64(11):1288-1298.( https://doi.org/10.1080/10962247.2014.943352)

[13]          Cornelis, G.; Etschmann, B.; Gerven, T.V.; Vandecasteenle, C. Mechanisms and modelling of antimonate leaching in hydrated cement paste suspensions[J]. Cement and Concrete Research, 2012,42(10): 1307-1316.( https://doi.org/10.1016/j.cemconres.2012.06.004)

[14]          Gao, W.; Li, Z.F.; Zhang, S.Q.; Zhang Y.Y.; Fu, P.F.; Yang H.F.; Ni, W. Enhancing Arsenic Solidification/Stabilisation Efficiency of Metallurgical Slag-Based Green Mining Fill and Its Structure Analysis[J]. metals, 2021,11: 1389.( https:// doi.org/10.3390/met11091389)

[15]          Gao, W.; Li, Z.F.; Zhang, S.Q.; Zhang Y.Y.; Teng, G.X.; Li, X.Q.; Ni, W.Solidification/Stabilization of Arsenic-Containing Tailings by Steel Slag-Based Binders with High Efficiency and Low Carbon Footprint[J]. materials, 2021,14: 5864.( https:// doi.org/10.3390/ma14195864)

[16]          Rasaki, S.A.; Zhang, B.X.; Guarecuco, R.; Thomas, T.J.; Yang, M.H. Geopolymer for use in heavy metals adsorption, and advanced oxidative processes: A critical review[J]. Journal of Cleaner Production, 2019,213: 42-58.( https://doi.org/10.1016/j.jclepro.2018.12.145)

[17]          Li, J.; Zhang, S.Q.; Wang, Q.; Ni, W.; Li, K.Q.; Fu, P.F.; Hu, W.T.; Li Z.F. Feasibility of using fly ash–slag-based binder for mine backfilling and its_associated leaching risks[J]. Journal of Hazardous Materials, 2020,400: 123191.

( https://doi.org/10.1016/j.jhazmat.2020.123191)

[18]          Li, Y.Y.; Ni, W.; Gao, W.; Zhang, Y.Y.; Yan, Q.H.; Zhang, S.Q. Corrosion evaluation of steel slag based on a leaching solution test[J]. Energy Sources, Part A: Recovery, Utilization, and Environmental Effects, 2019,41(7): 790-801.

( https://doi.org/10.1080/15567036.2018.1520359)

[19]          Cornelis, G.; Gerven, T.V.; Vandecasteele, C. Antimony leaching from MSWI bottom ash Modelling of the effect ofpH and carbonation[J]. Waste management, 2012,32(2): 278-286.( https://doi.org/10.1016/j.wasman.2011.09.018)

[20]          Cornelis, G.; Johnson, C.A.; Gerven, T.V.; Vandecasteele, C. Leaching mechanisms of oxyanionic metalloid and metal species in alkaline solid wastes[J]. Applied Geochemistry, 2008,23(5): 955-976.( https://doi.org/10.1016/j.apgeochem.2008.02.001)

[21]          Gao, W.; Ni, W.; Zhang, Y.Y.; Li, Y.Y.; Shi, T.Y.; Li, Z.F. Investigation into the semi-dynamic leaching characteristics of arsenic and antimony from solidified/stabilized tailings using metallurgical slag-based binders[J]. Journal of Hazardous Materials, 2020,381: 120992.( https://doi.org/10.1016/j.jhazmat.2019.120992)

[22]          Zhang, Y.Y.; Zhang, S.Q.; Ni, W.; Yan, Q.H.; Gao, W.; Li, Y.Y. Immobilisation of high-arsenic-containing tailings by using metallurgical slag-cementing materials[J]. Chemosphere, 2019,223:117-123.( https://doi.org/10.1016/j.chemosphere.2019.02.030)

[23]          Zhang, Y.Y.; Gao, W.; Ni, W.; Zhang, S.Q.; Li, Y.Y.; Wang K.; Huang, X.H.; Fu, P.F.; Hu W.T. Influence of calcium hydroxide addition on arsenic leaching and solidification/stabilisation behaviour of metallurgical-slag-based green mining fill[J]. Journal of Hazardous Materials, 2020,390: 122161.

( https://doi.org/10.1016/j.jhazmat.2020.122161)

[24]          Yan, Q.H.; Ni, W.; Gao, W.; Li, Y.Y.; Zhang, Y.Y. Mechanism for solidification of arsenic with blast furnace slag-steel slag based cementitious materials[J]. Journal of Central South University(Science and Technology), 2019,50(07): 1544-1550.

( https://doi.org/10.11817/j.issn.1672-7207.2019.07.006)

[25]         Li, Y.C.; Wu, J.X.; Hu, W.; Ren, B.Z.; Hursthouse, A.S. A mechanistic analysis of the influence of iron-oxidizing bacteria on antimony (V) removal from water by microscale zero-valent iron[J]. Journal of Chemical Technology & Biotechnology, 2018,93.(https://doi.org/10.1002/jctb.5606)

[26]          Yi, H.; Xu, G.P.; Cheng, H.G.; Wang, J.S.; Wan, Y.F.; Chen, H. An Overview of Utilization of Steel Slag[J]. Procedia Environmental Sciences, 2012,16: 791-801.( https://doi.org/10.1016/j.proenv.2012.10.108)

[27]          Li, Y.;  Qiao, C.y.; Ni W. Green concrete with ground granulated blast-furnace slag activated by desulfurization gypsum and electric arc furnace reducing slag[J]. Journal of Cleaner Production, 2020,269: 122212.( https://doi.org/10.1016/j.jclepro.2020.122212)

[28]          Liu, Y.; Ni, W.; Huang, X.Y.; Ma, X.M.; Li, D.Z. Characteristics of hydration and hardening of red mud of bayer process in carbide slag-flue gas desulfurization gypsum system[J]. Materials Reports, 2016,30(14): 120-124.

(https://doi.org/10.11896/j.issn.1005-023X.2016.14.027)

[29]          Ni, W.; Li, Y.; Xu, D.; Xu, C.W.; Jiang, Y.Q.; Gao, G.J. Hydration mechanism of blast furnace slag-reduction slag based solid waste cementing materials[J]. Journal of Central South University(Science and Technology), 2019,50(10): 2342-2351.

( https://doi.org/10.11817/j.issn.1672-7207.2019.10.002)

[30]          Eskander, S.B.; Bayoumi, T.A.; Tawfik, M.E. Immobilization of Borate Waste Simulate in Cement-Water Extended Polyester Composite Based on Poly(Ethylene Terephthalate) Waste [J]. Polymer-Plastics Technology and Engineering, 2006,45(8): 939-945.

( https://doi.org/10.1080/03602550600723415)

[31]          Zhao, Y,L.; Wu, P.Q.; Qiu J.P.; Guo, Z.B.; Tian, Y,S.; Sun, X,G,; Gu, X.W. Recycling hazardous steel slag after thermal treatment to produce a binder for cemented paste backfill[J]. Powder Technology, 2022,395: 652-662.

( https://doi.org/10.1016/j.powtec.2021.10.008)

[32]          Ines, G.L.; Fernandez-Jimenez A.; Blanco, M.T.; Palomo, A. FTIR study of the sol–gel synthesis of cementitious gels: C–S–H and N–A–S–H[J]. Journal of Sol-Gel Science and Technology, 2008,45(1): 63-72.( https://doi.org/10.1007/s10971-007-1643-6)

[33]          Noam, A. Covalent radii from ionization energies of isoelectronic series[J]. Chemical Physics Letters, 2014,595-596: 214-219.( https://doi.org/10.1016/j.cplett.2014.01.037)

[34]     Cornelis, G.; Johnson, C.A.; Gerven, T.V.; Vandecasteenle, C. Leaching mechanisms of oxyanionic metalloid and metal species in alkaline solid wastes: A review[J]. Applied Geochemistry, 2008,23(5): 955-976.( https://doi.org/10.1016/j.apgeochem.2008.02.001)

[35]          Santamaria, L.; Vicente, M.A.; Korili, S.A.; Gil, A. Effect of the preparation method and metal content on the synthesis of metal modified titanium oxide used for the removal of salicylic acid under UV light[J]. Environmental Technology, 2018,41: 2073-2084.( https://doi.org/10.1080/09593330.2018.1555285)

[36]          Li, Y.; Wu, B.H.; Ni, W.; Li, X.M. Synergies in Early Hydration Reaction of Slag-Steel Slag-Gypsum System[J]. Journal of Northeastern University( Natural Science), 2020,41(4): 581-586.( https://doi.org/10.12068/j.issn.1005-3026.2020.04.022)

[37]          Avalos, N.M.; Varga, T.; Mergelsberg, S.T.; Silverstein, J.A.; Saslow, A.A. Behavior of iodate substituted ettringite during aqueous leaching[J]. Applied Geochemistry, 2021,125: 104863.( https://doi.org/10.1016/j.apgeochem.2020.104863)

[38]          Pointeau, I.; Reiller, P.; Mace, N.; Landesman, C.; Coreau. N. Measurement and modeling of the surface potential evolution of hydrated cement pastes as a function of degradation[J]. Journal of Colloid & Interface Science, 2006,300(1): 33-44.( https://doi.org/10.1016/j.jcis.2006.03.018)

[39]          Zou, C.; Long, G.C.; Zheng, X.H.; Ma, C.; Xie, Y.J.; Sun, Z.P. Water evolution and hydration kinetics of cement paste under steam-curing condition based on low-field NMR method[J]. Construction and Building Materials, 2020,271: 121583.( https://doi.org/10.1016/j.conbuildmat.2020.121583)

[40]          Elakneswaran, Y; Nawa, T.; Kurumisawa, K. Electrokinetic potential of hydrated cement in relation to adsorption of chlorides[J]. Cement & Concrete Research, 2009,39(4): 340-344.( https://doi.org/10.1016/j.cemconres.2009.01.006)

[41]          Ochs, M.; Pointeau, I.; Giffaut, E. Caesium sorption by hydrated cement as a function of degradation state: Experiments and modelling[J]. Waste Management, 2006,26(7): 725-732.( https://doi.org/10.1016/j.wasman.2006.01.033)

[42]          Cornelis, G.; Gerven, T.V.; Snellings, R.; Verbinnen, B.; Elsen, J.; Vandecasteele, C. Stability of pyrochlores in alkaline matrices Solubility of calcium antimonate[J]. Applied Geochemistry.2011,26(5): 809-817.( https://doi.org/10.1016/j.apgeochem.2011.02.002)

Round 2

Reviewer 2 Report

Dear authors,

In my opinion, your manuscript was significantly improved after revision, and I received detailed answers to all my comments.